# Citrus Canker Pathogen, Its Mechanism of Infection, Eradication, and Impacts

**DOI:** 10.3390/plants12010123

**Published:** 2022-12-26

**Authors:** Esha Shahbaz, Mobeen Ali, Muhammad Shafiq, Muhammad Atiq, Mujahid Hussain, Rashad Mukhtar Balal, Ali Sarkhosh, Fernando Alferez, Saleha Sadiq, Muhammad Adnan Shahid

**Affiliations:** 1Department of Food Sciences, Faculty of Agricultural Sciences, University of the Punjab, Lahore 54590, Pakistan; 2Department of Horticulture, Faculty of Agricultural Sciences, University of the Punjab, Lahore 54590, Pakistan; 3Department of Plant Pathology, University of Agriculture, Faisalabad 38000, Pakistan; 4Horticultural Science Department, North Florida Research and Education Center, University of Florida/IFAS, Quincy, FL 32351, USA; 5Department of Horticulture, College of Agriculture, University of Sargodha, Sargodha 40100, Pakistan; 6Horticultural Sciences Department, University of Florida, Gainesville, FL 32611, USA; 7Horticultural Science Department, Southwest Florida Research and Education Center, University of Florida/IFAS, Immokalee, FL 34142, USA

**Keywords:** citrus canker, epidemiology, eradication, future prospects, pathogen profile, symptoms, *Xanthomonas citri* pv. *citri*, *Xanthomonas citri* pv. *aurantifolii*

## Abstract

Citrus canker is a ravaging bacterial disease threatening citrus crops. Its major types are Asiatic Canker, Cancrosis B, and Cancrosis C, caused by *Xanthomonas citri* pv. *citri* (Xcc), *Xanthomonas citri* pv. *aurantifolii* pathotype-B (XauB), and pathotype-C (XauC), respectively. The bacterium enters its host through stomata and wounds, from which it invades the intercellular spaces in the apoplast. It produces erumpent corky necrotic lesions often surrounded by a chlorotic halo on the leaves, young stems, and fruits, which causes dark spots, defoliation, reduced photosynthetic rate, rupture of leaf epidermis, dieback, and premature fruit drop in severe cases. Its main pathogenicity determinant gene is *pthA*, whose variants are present in all citrus canker-causing pathogens. Countries where citrus canker is not endemic adopt different methods to prevent the introduction of the pathogen into the region, eradicate the pathogen, and minimize its dissemination, whereas endemic regions require an integrated management program to control the disease. The main aim of the present manuscript is to shed light on the pathogen profile, its mechanism of infection, and fruitful strategies for disease management. Although an adequate method to completely eradicate citrus canker has not been introduced so far, many new methods are under research to abate the disease.

## 1. Introduction

One of the major threats to sustainable crop production and food security is the increased number of invasive and aggressive plant pathogens. They greatly decrease the yield and quality of the crop. Citrus forms the backbone of the agriculture industry in many countries [1]. Its growth and yield are affected by many pathogens. Citrus canker, one of the most ravaging biotic stresses to citrus, causes substantial economic impacts to the citrus industry, thus limiting trade and production. It affects all commercial citrus varieties and many related rutaceous species [2]. This review aims to shed light upon advanced research studies regarding the pathogen profile of citrus canker, its mechanism of infection, and management strategies; provide an overview of previous research studies; and highlight the sectors that may need more attention to find success in complete eradication of this disease.

## 2. Historical Perspective

The region where citrus canker originated is a matter of disagreement. It is suspected that it emerged from Southeast Asia [3,4]. Because of the identification of canker lesions on the oldest citrus herbaria in England’s royal botanical garden, it was reported that citrus canker first emerged in India and Indonesia. From 1827–1831, infected *Citrus medica* was collected from India. From 1842–1844, infected *Citrus aurantifolia* was collected from Indonesia. [5,6]. In other research, Lee (1918) suspected southern China to be the origin of citrus canker A [7,8]. The geographical origin of citrus canker is a controversial topic. The reports lead us to believe that citrus canker disease originated from tropical areas in Asia such as India, Indonesia, and China.

It is believed that citrus canker spread to other regions largely by the movement of budwood. Eventually, this disease was traced in the Gulf States region of the US in 1915. The shipment of contaminated nursery stock from Asia is suspected to be responsible for the eruption in the Gulf States [9]. The presence of this disease has also been reported in other countries such as Argentina, Brazil, Australia, Oman, Saudi Arabia, and Uruguay. Its spread affected Asia, South America [10], South Africa [11], Oceania, and Australia [12].

Citrus canker has an extensive history in Florida. The pandemics that occurred in Mexico in 1981 and Florida in 1984 were non-identical to the Asiatic strain of canker [9,13]. A large-scale epidemic was also found in urban Miami, Florida in 1995. In some of these locations, efforts were made to remove the bacteria causing disease, but not all efforts were successful. It was rediscovered in 1997, and the efforts to remove this disease began again [2].

Keeping in view the significant losses caused by citrus canker, efforts to eradicate it and prevent further spreading were made in all the previously mentioned countries. These methods consisted of ignition of the on-site trees, application of bactericidal formulations, routine checkups of the plant nurseries and orchards, and isolation. Some regions such as Australia and Argentina [14] have been successful in eradicating the pathogen, whereas many others including Brazil, Mali, Uruguay, and Florida are still struggling for disease management [13].

Despite the efforts to eradicate the citrus canker in America, New Zealand, England, and South Africa, it reappeared again in some countries. After approximately 70 years of successful eradication, the Asiatic canker strain was again reported in Texas in May 2016 [15]. Similarly, after successful eradication of citrus canker from South Africa during the 20th century, it reappeared in 2006 in several African countries including Mali, Somalia, Ethiopia, and Senegal [16].

Eradication programs for citrus canker require extensive budgets, which led to their termination in some countries/states that were continuously getting exposed to citrus canker, such as Florida (program terminated in 2006) and Brazil (terminated in 2009) [17]. Management practices and other regulations are still implemented by growers in such areas to limit the losses. In Pakistan, the first case of citrus canker originated from Punjab. Citrus canker is found mostly in the areas of Punjab where citrus is grown in a commercial setting. [18]. In India, this disease was first reported in Punjab [19,20]. It was also found in Tamil Nadu. The disease is a grave issue for acid lime (*C. aurantifolia*) in places where it is being grown in a large setting for market purposes (e.g., Akola area in Central India, Nellore and Periyakulum regions in Southern India, and Khera region of Western India) [21]. Later, it was also discovered in Kinnow Mandarin nurseries in Punjab [9].

## 3. Pathogen Profile

The causative agents of citrus canker are *Xanthomonas citri* pv. *Citri*, *Xanthomonas citri* pv. *Aurantifolii,* and their pathotypes. Citrus canker is a rod-shaped, slender, gram-negative bacterium belonging to the family Xanthomonadaceae, which is considered as one of the most significant and largest families of bacterial plant pathogens. It is capable of producing slow-growing, non-mucoid colonies in culture [2,22]. It usually ranges in size from 1.5–2.0 × 0.5–0.75 μm [22]. The taxonomy of its causative agents and their strains has always been problematic [23,24,25,26,27]. Previous investigations revealed that the first research that focused on the description of citrus canker was carried out in 1914 by the Bureau of Plant Industry, United States Department of Agriculture (based on the samples from orange and grapefruit growers in Florida, Texas, and Mississippi). It was then declared a new organism, named ‘*Pseudomonas citri*’, as its etiological agent. It was described as a short, motile rod bacterium with rounded ends along with a polar flagellum [28,29]. It was also reclassified as *Xanthomonas campestris* pv. *citri* and then as *Xanthomonas axonopodis* pv. *citri* after its detailed genomic investigation [23,25,27]. Similarly, the name of the causative agent of cancrosis B and C is usually published as *Xanthomonas fuscans* pv *aurantifolii* or *Xanthomonas citri* pv *aurantifolii* [30,31]. Here, we are abbreviating the causative agents of canker A, B, and C as Xcc, XauB, and XauC, respectively. Stable classification of the species of genus ‘*Xanthomonas*’ has been a challenge for a long time, and the reason for it is that this genus possesses a great phytopathogenic diversity in contrast to a phenotypic uniformity [32]. Currently, the most acceptable and validly published name of the causative agent of citrus canker is *Xanthomonas citri*, whose different pathovars are responsible for causing different types of citrus canker [27]. It is regarded as a quarantine organism in various citrus-producing countries that are canker-free or have successfully eradicated the disease, in order to limit its spread [33].

### Major Types of Pathovars of Xanthomonas citri

There are five major pathovars belonging to *Xanthomonas citri* that are responsible for causing diseases in citrus, i.e., *citri*, *aurantifolii* (Pathotype B, C, D) and *citrumelo* (also referred to as type E), and they also have many bacterial strains [25,34,35]. These five pathovars were initially studied as causal agents of citrus canker A, B, C, D, and E, respectively, but detailed studies have highlighted a controversy about the causative agents of citrus canker D and E.

The identification of pathotype-D was controversial, as only one pathogenic bacterial strain of this pathotype had been isolated and identified. Disease caused by it was commonly referred to as Citrus bacteriosis. Later, it was named Citrus leaf spot (*mancha foliar de los citricos*), and the fungus *Alternaria limicola* was confirmed as its actual causative agent [35]. The symptoms caused by E-strain, as well as its host range, genome, pathogenicity, and many other factors varied significantly from characteristics of other canker-causing pathovars of *Xanthomonas citri,* which led to the reclassification of citrus canker E (Florida nursery form of citrus canker) as Citrus Bacterial Spot (CBS) disease [36,37,38,39]. These pathovars were found to be associated with each other at the levels of DNA binding of greater than 60%, which reveals that they belong to one species [40]. They can be distinguished from each other using various physiological techniques, serological tests, fatty acid analyses, pathogenicity tests, PCR-based assays, total protein files, RFLP analyses, genome sequencing, restriction enzyme analyses of amplified DNA fragments of an *HRP*-related DNA sequence, DNA–DNA hybridization, or any other biochemical or molecular technique [41,42,43,44,45,46,47,48]. Table 1 shows a brief comparison between these five pathovars.

## 4. Types of Citrus Bacterial Canker

Three major types of citrus bacterial canker (CBC) are known, which are named Asiatic canker (citrus canker A), cancrosis B (citrus canker B), and cancrosis C (citrus canker C) caused by *Xanthomonas citri* pv. *citri* (Xcc), *Xanthomonas citri* pv. *Aurantifolii* pathotype-B (XauB), and pathotype-C (XauC), respectively [31,49]. The most aggressive one is Asiatic canker (citrus canker A) [9,50].

Different phenotypic traits of the three strains have been compared, and their phylogenetic relationships and comparative genomic analyses have been studied in detail in various studies. Different molecular and genetic analyses have shown that XauB and XauC strains are more closely associated with each other as compared to the Xcc strain. They all elicit almost similar symptoms but vary greatly in aggressiveness, host range, and geographical distribution (mentioned in detail in Table 1). The three strain types contain 65 families of orthologous genes that are specific to them and are absent in all other completely sequenced species of genera *Xanthomonas* and *Xylella*. The Xcc-specific genes are greater in number than the analogous genes for XauB and XauC [30]. The three strains can be robustly classified into two distinct clades, i.e., C-c clade or citri-citri clade, consisting of all A strains, and aurantifolii clade, consisting of XauB and XauC strains [31].

Table 2 shows a brief comparison of their various characteristics, genetic analyses, and responses to some basic identification tests. Analyzing genomes of the strains comparatively allows us to understand the difference in their virulence, host ranges, and metabolism. For instance, the absence of genes encoding for hlyD (an ABC transporter) and hlyB (a membrane-fusion protein) in the type 1 secretion system (T1SS) of XauB strains, which are involved in toxin secretion, might contribute to their decreased competitive capability with other organisms. Similarly, the lack of specific gene clusters from the type 4 secretion system (T4SS) in XauB and XauC strains, which play a dominant role in bacterial pathogenesis, adaptation, and cellular interaction, might play a role in their competitive capability with other bacteria. The absence of some specific gene clusters in the type 4 pilus (T4p) and genes involved in hemagglutinin and hemolysin synthesis in XauB and XauC strains affect their self-aggregation capability, tissue adhesion process, and biofilm formation. The absence of vapBC and tspO genes is suspected to be one of the reasons behind decreased pathogenicity of XauB and XauC strains. Some effectors (xopF1, xopB, xopE4, xopJ, xopAF, and xopAG) which are absent in Xcc strains might be the cause of host range restriction in XauB and XauC strains. Presence of the xacPNP gene only in Xcc strains contributes to its high virulence as compared to other strains [30,31,51]. All the strains contain at least one pthA or pthA-like gene, which is considered as a crucial pathogenicity determinant. Its presence is necessary to elicit canker lesions and plays a noteworthy role in restricting the host range of these Xanthomonas species to citrus species and some closely related rutaceous species [52,53]. Effectors xopA1 and xopE3 which are present in all strains of citrus canker are also suspected to play a significant role in citrus canker [30].

### Pathogenic Variants of Citri Pathovar

Two variants of *Xanthomonas citri* pv. *citri* were reported later. The pathotype A* was identified for the first time in Mexican limes (*Citrus aurantifolii*) in various Southwest or Western Asian countries including Oman, Saudi Arabia, Iraq, UAE, and Iran in the 1990s [39]. Pathotype $A^{w}$ (‘w’ represents ‘Wellington’ strain) was identified for the first time in Mexican limes (*Citrus aurantifolii*) and alemow (*Citrus macrophylla* Wester) trees situated in Southern Florida (Wellington and Lake Worth) in 2000. However, other susceptible citrus trees surrounding the infected trees did not exhibit any canker symptoms, thus indicating the host specificity of this pathotype [34]. Afterwards, it was reported that these strains probably had a common origin in India (Southwest Asia), which is also the putative origin of pathotype A [37,55].

The three pathogenic variants A, A*, and $A^{w}$ mainly differ in pathogenicity and host ranges. The pathotype A has a broad host range, whereas the other two have restricted host ranges, possessing the ability to elicit canker lesions only on Mexican/key lime and alemow under natural conditions. They are also geographically restricted, whereas pathotype A is found in several citrus-producing regions [34,39]. They were initially distinguished from pathotype A due to their inability to produce typical canker lesions on Duncan grapefruit, which is highly susceptible to Xcc pathotype A [56]. Pathotype $\;A^{w}$ varies from pathotype A* due to its ability to induce a hypersensitive response (HR) in grapefruit, sweet orange, and some other citrus species [34]. *XopAG* (*avrGf1*) is considered to be responsible for inducing the hypersensitive response (HR), which is found in $A^{w}$ strains, thus acting as a host-restricting factor. It is also found in very few A* strains.

The three pathogenic variants vary in the presence or absence of genes belonging to the classes of the secretion system, effectors, lipopolysaccharides, and other functional groups, which may influence the variation in their virulence, host range specificity, metabolism, etc. [57]. They greatly exhibit genomic differences related to horizontal gene transfer, single nucleotide polymorphism, and recombination [48]. It is suggested that the acquisition of productive genes and lack of detrimental genes is most probably the reason behind the ability of pathotype A to attack a broader host range than A* and $A^{w}$ pathotypes [58]. Many types of research have been conducted to study their comparative genomic analysis and evolutionary history in detail [42,48,51,59,60,61].

These pathogenic variants were associated with pathovar *citri* due to their genetic and phenotypic correspondence with pathotype A [56]. DNA reassociation analysis revealed that all Xcc-$A^{w},$ Xcc-$A^{*},$ and Xcc-A strains are closely related, and their DNA similarities range from 70.7–94.1%. Their DNA similarities with the strains of *X. citri* pv. *aurantifolii* range from 34.6 to 50.6%, which represents that they are quite dissimilar [34]. A*/$A^{w}$ strains show greater genetic diversity than the A-strain [59]. At the pathotype level, A* strains are the most diverse, having the highest average genome polymorphism [48].

## 5. Symptoms

All the newly growing parts of a plant are vulnerable to this bacterium. The symptoms of citrus canker include noticeable necrotic lesions that appear on leaves, twigs, and fruits. These early symptoms show as early as 4–7 days after the entry of bacterium into the plant tissues under ideal conditions, which include the presence of water film and temperatures in the range of 20–30 °C [62]. If the conditions are not ideal, the symptoms take longer than expected, i.e., more than 60 days [63]. The lesions can be felt by moving the fingers on the surface of the affected group of cells. Their center then becomes lifted and cork-like [9].

Xcc has the ability to normally affect green citrus tissues when they are in the expansion state of growth. The leaves, twigs, and fruits become more immune to injury when they reach their full size of maturity and start to harden off [13]. An important indication of citrus canker is citrus tissue hyperplasia (more than usual mitotic cell division), resulting in the elicitation of canker lesions [9,52].

### 5.1. On Leaves

In the case of leaves, the first symptoms that appear are 2–10 mm erumpent circular patches usually on the lower epidermis or the side facing away from the stem [9]. Lesions appear mostly on both sides of the leaf. On leaves, round spots are lifted and form vesicle-like soft eruptions that could either be white or yellow. These eruptions then become stiff and become a light tan. Afterwards, they evolve into corky canker lesions that do not feel smooth when touched. Frequently, a water-soaked margin is created around the necrotic tissue [37]. In due course, a depression is formed in the center of the lesions of the leaf, eventually leading them to fall out. This creates a shot-hole effect [55]. Premature removal of leaves and the progressive death of twigs pose the major problems for the plant as the disease escalates. Most lesions on a leaf are of similar size because there is typically only one infection period.

### 5.2. On Twigs and Fruits

In the case of twigs and fruits, the indications of citrus canker are almost the same. These symptoms include elevated lesions along with a boundary of oil or a water-soaked margin. Commonly, there is no chlorosis around twig lesions, but there is a chance that it might occur in fruit lesions. The indications of chlorosis take time to dissipate. In the areas where citrus canker is domestic, twig wounds present on the pointed new shoots supply most of the sustaining Xcc infusion [13]. Lesions present on stems may collapse to divide the epidermis along the stem length, and banding of immature stems may happen most frequently [9].

The untimely falling and discoloration of fruit are the major economic effects of this disease. The fruits with cankers are rejected by the fresh trade, which causes substantial economic loss. The lesions on leaves and stems that are elderly have more raised edges, and most of the time, they are bounded by a yellow chlorotic disc (that might vanish as the canker lesions become old) and a sunken center. Sunken centers are most of the time detectable on fruits, but the lesions do not perforate deep into the outer skin of the fruit and do not affect the inside quality [9,55].

## 6. Pathogen’s Infection Mechanisms

The Xcc bacteria attack stems, leaves, and fruits, with the major entrance sites being stomata and wounds. After the bacteria colonize the apoplast, the pathogen that caused cell hyperplasia damages the leaf epidermis. The lesions on stems, leaves, and fruits rise, darken, and become thick, forming the distinctive elevated necrotic corky lesion. The pathogen can be transferred to new growth on other citrus plants after the disease propagates in lesions and in the presence of moisture on them [64]. The genes that code for bacterial attachment and shallow structure components, protein secretion systems, poisons, and plant cell wall-degrading enzymes have all been discovered in Xcc and help it to survive and cause disease in citrus [65]. A study suggests that Xcc may use several pathways to attack its host [66]. Figure 1 briefly explains the disease cycle of citrus canker.

### 6.1. Role of Adhesin Proteins in Bacterial Adhesion

In the pathogenicity phase, bacterial adhesion is very critical. It is mediated by adhesin proteins that are secreted from the type V protein secretion system. It consists of two different routes that translocate protein domains or large proteins. One is the autotransporter pathway, whereas the other one is the TPS (two-partner secretion) pathway [67]. At least one homolog to the TPS type V secretion system is present in the Xcc genome, whose function is to secrete an FhaB-like hemagglutinin protein that codifies for the XacFhaB protein. The gene XacFhaC, which is located upstream of XacFhaB, encodes a putative TpsB partner secretion of the XacFhaB protein [68]. Mutation in XacFhaB disrupts bacterial adhesion and aggregation and causes a more dispersed, smaller, and decreased number of cankers. Mutants in the transporter protein XacFhaC, on the other hand, exhibited an intermediate virulence phenotype similar to the infection of wild-type bacteria, which suggests that changing to that XacFhaB could also be released from another partner other than XacFhaC [69]. Thus, this adhesin protein is critical in the early phases of infection [66].

### 6.2. Significance of Type III Protein Secretion System

Invasion of the pathogen into the host’s cell triggers its immunity system. After successfully landing on a host, the pathogen must pass through the host’s cell wall to reach the cell membrane, where they interact with the receptors that detect PAMPs (pathogen-associated molecular patterns). When the pathogen is identified by these receptors, it triggers PAMPs-mediated immunity that eliminates the pathogen. In order to deal with it, the pathogen possesses the ability to intervene with the identification in the cell membrane, or it secretes effector proteins that may change the resistance response expression into the plant’s cytosol [66]. Flagellin of Xcc flagella is a common PAMP that triggers the plant’s immunity [70].

When the pathogens successfully cancel out the primary immunity of plants, they encounter a more specific pathogen detection system developed by plants that includes direct or indirect identifying of microbial proteins by resistance proteins (R) and activating the resistance signaling routes [71].

The type III protein secretion system is crucial for pathogenicity and is codified by the HRP (for hypersensitive response and pathogenicity) cluster [72,73]. Xcc induces a hypersensitive reaction in resistant plants and non-host plants [74]. A T3SS consists of a flagellum (in phytopathogens, the HRP pilus) and a basal body that maintains the stability of the structure. This complicated structure facilitates bacterial adhesion to the host cell membrane to guarantee that effector proteins are transported to the plant cell’s internal region [75]. This system secretes many effector proteins [76]. PthA, the crucial pathogenicity determinant of citrus canker, is also secreted by this system, and it belongs to the *avrBs3* gene family [52]. Its expression is enough to cause the death of host plant cells [77]. The HRP cluster is crucial for the occurrence of HR in non-host and citrus canker in host plants [78].

### 6.3. Xanthan and Biofilm Formation

Biofilm formation plays a crucial role in the bonding of the pathogen and its host. Thus, it is a necessary factor for a successful pathogen attack, survival, and epiphytic fitness, but it does not affect the virulence of the pathogen [79,80]. It is made of xanthan polysaccharide and some other components [22]. The genus Xanthomonas is distinguished by the production and emission of an exopolysaccharide (EPS) xanthan, which is manufactured by FhaB and the gum operon [81]. It is modulated by the *rpf* (regulation of pathogenicity factors) gene cluster [82] which also encodes the cell–cell signaling system [83]. *GumD*, a crucial component of biofilm formation, is a glycotransferase gene of gum cluster and catalyzes the initial step involved in the production of xanthan [79]. *GumD* mutants showed reduced oxidative stress survival and epiphytic survival on citrus leaves during the stationary phase, which shows that xanthan facilitates the survival of bacteria on the host plant [80]. The XacFhaB adhesin as well as the EPS take part in the production of biofilms in Xcc. These molecules are also linked to bacterial motility, which results in bigger cankers on citrus leaves infected with the bacterium [69]. When an adequate population of the pathogen manages to invade the host cell, they shed their flagella and aggregate to form biofilm. Stages of biofilm formation are shown in Figure 2.

### 6.4. Damage to Host’s Machinery

Xcc contains a plant natriuretic peptide (PNP)-like gene known as XacPNP that is produced during citrus canker disease, but no other phytopathogen or bacterium does [84,85]. Higher plants have been found to rely on natriuretic peptides (PNPs) to regulate their salt and water balance and to grow [86]. They have been found in conductive tissue [87] and become active in response to osmotic stress and $K^{+}$ deficiency [88]. XacPNP has a lot of similarities to PNPs in terms of its sequence and domain structure, and it can trigger physiological responses in plants such as stomatal opening [85]. Lesions generated by citrus canker in leaves infected by a xacPNP mutant are more necrotic than those produced by wild-type bacteria, which eventually results in early bacterial cell death [85,89]. This suggests that Xcc creates favorable conditions for its survival in the plant’s cell by modifying its responses [85,89].

Moreover, Xcc also decreases the photosynthesis efficiency of host plant cells. Decreases in the expression of sugar-regulating proteins including Rubisco and Rubisco activase, as well as of ATP synthase and an increase in NADH dehydrogenase were observed, indicating a loss in efficiency of photosynthesis [90]. It also increases the synthesis and movement of gibberellic acid, auxins, and ethylene in the host cell [91,92]. PthA and PthA2 of T3SS regulate the auxin and gibberellin synthesis and assign RNA polymerase II to start its targeted transcription. They also regulate the organization of the cell wall, the transport of lipids, and the sugar metabolism of host cells [92].

## 7. Eradication and Control Measures

The increase in international travel and trade has dramatically escalated the risk of introducing invasive plant pests and pathogens to crops [37]. In order to prevent the introduction of and limit the spread of disease and eradicate it, various methods are being adopted. A proper eradication technique for citrus canker has not been developed until now, but various measures could be taken to control it. It has been a center of focus for researchers since its discovery [2]. Countries where this disease is not endemic rely on adopting different methods to avoid the introduction of the bacterial pathogen into the region, reduce the inoculum sources, eradicate the pathogen, and minimization its dissemination [93,94]. The regions where citrus bacterial canker is endemic require an integrated management program for its control which includes planting resistant varieties and canker-free nursery stock, establishing windbreaks and fences, spraying copper bactericides, disinfecting, controlling attacks of citrus leaf miner, and applying systematic acquired resistance (SAR) inducers [37,94,95]. Some of the efficacious methods of citrus canker control are mentioned below.

### 7.1. Physical Approaches

#### 7.1.1. Tree Removal

Tree removal is efficacious only when the disease is localized and limited to a small number of citrus trees. Previously, the “rule of 1900 feet” was used to eradicate the infected trees. According to that rule, all trees within a radius of 1900 feet (579 m) from an infected tree were removed to limit the spread of the disease [55]. The trees were then reduced to pieces of generally less than 10 cm in size through wood chipping machinery and were dumped and covered with soil [2]. Replantation of citrus in the areas that had undergone the eradication process was allowed only when they were declared canker-free for at least 2 years [96]. This regulation was included in Florida’s Citrus Canker Eradication Program (CCEP) and practiced at the end of 1999 in Florida. It gained negative responses from commercial and residential citrus growers [96,97]. This “1900 feet rule” was suspended in January 2006 as it was not an economically sustainable method [98].

#### 7.1.2. Periodic Inspection of Citrus Orchards

After some interval of time, the regular inspection of citrus trees facilitates quick identification of the pathogen; thus, it helps in rapid eradication of the disease.

#### 7.1.3. Windbreaks

The spread of citrus canker is exacerbated by warm, humid, cloudy climates, strong winds, and heavy rainfall. Establishing windbreaks help in the prevention of direct interactions of wind with citrus trees, thus reducing penetrations of the pathogen into the host. Moreover, strong winds may damage the tissues of citrus trees, which causes wounds on the plant surface and facilitates the bacterial attack, but this can be reduced by establishing windbreaks. Establishing windbreaks in combination with the application of copper bactericides has been proven effective for controlling citrus canker [99].

#### 7.1.4. Planting Resistant Varieties

Different citrus cultivars vary in susceptibility to citrus canker. Planting resistant cultivars in the region where the risk of spread of the pathogen is relatively high can help in minimizing the economic losses caused by citrus canker. The more canker-resistant varieties are kumquat (*Fortunella* spp.), calamondin (*Citrus mitus*), and citron (*Citrus medica*) [9,37]. The practicality of this strategy is limited given the limited commercial potential of those varieties.

#### 7.1.5. Pruning or Defoliation

Pruning or defoliation of infected summer and autumn shoots before a monsoon and burning is beneficial to control canker spread. Moreover, the application of copper sprays in combination with pruning has proven to be significantly beneficial in controlling minor outbreaks of citrus canker [9,37].

#### 7.1.6. Other Precautionary Measures

Other precautionary measures, such as the sanitization of instruments and workers and the careful use of instruments to avoid wounds, should also be observed in citrus orchards and nurseries [9].

### 7.2. Chemical Approaches

#### 7.2.1. Copper Sprays

Different experiments have proven that the application of copper products for the control of citrus canker is quite efficacious. Copper bactericides can diminish the bacterial population present on the leaf’s surface. They develop a protective layer over the leaves and the fruits that disappears with the expansion of the surface area. They need to be sprayed multiple times after some interval of time to achieve sufficient control on susceptible hosts [2,100]. The most suitable time for its application is during the summer and spring months when the conditions are most favorable for pathogen attacks [101].

Treatment with the sprays of copper ammonium carbonate (CAC), copper hydroxide, copper pentahydrate, copper oxychloride along with kasugamycin, tribasic copper sulfate, and the mixture of sodium arsenate and copper sulfate was also found effective for controlling citrus canker through different experiments [95,102,103]. Magna-bon is a chelated copper formulation whose treatment has also been proven beneficial for the control of this disease and has a low metallic copper ratio (5%) [95,104].

The efficacy of copper sprays is adversely affected by wind-driven rain that directly introduces the pathogen into stomata, bypassing the protective layer of copper developed on the surface, but this can be overcome with the help of windbreaks [105,106]. Moreover, the continuous application of copper sprays for a long time induces resistance to copper in the pathogen population [2], which can be overcome with the addition of maneb or mancozeb to the copper bactericides [9,102,107]. Another major disadvantage of using copper products is that their continuous application can cause accumulation of copper in citrus soils that can be disadvantageous for the soil, the environment, and the plant [108]. In order to lessen its unhealthy effects, it can be used in combination with some other agent (such as streptomycin) with a lower metallic copper rate [95]. Tree-Row-Volume (TRV) methodology can also be helpful in this regard as it reduces the usage of water and chemical in sprays and is suitable for reducing citrus canker through copper sprays [109].

#### 7.2.2. Alternative Bactericides

Zinkicide is a zinc-oxide-based nano-formulation that has the potential to limit the spread of citrus canker and reduce the development of canker lesions. It is easier to spray, has high anti-microbial activity, and is less toxic compared to copper and copper-zinc formulations [110]. Another formulation consisting of didecyldimethylammonium and zinc-chelate, named MS3T (multifunctional surface, subsurface and systematic therapeutic) formulation, not only controls citrus canker but also improves fruit yield and quality [111]. Similarly, hexanoic acid [112] and hexyl gallate (G6) can also be used as an alternative to copper sprays. Hexyl gallate targets the outer membrane of the pathogen and can be used long-term without inducing resistance in the bacteria [113].

#### 7.2.3. Application of Antibiotics

Antibiotics represent one of the main discoveries of the last century that changed the treatment of a large array of infections in a significant way. However, increased consumption has led to an exposure of bacterial communities and ecosystems to large amounts of antibiotic residues. The application of various antibiotics (such as gentamicin and streptomycin) hampers the growth of the pathogen but is less effective than copper sprays and may induce resistance in the pathogen when used excessively [114,115]. Currently, there are concerns over resistance, but it has not led to any legal measures to reduce antibiotic use. However, antibiotic use in communities has been reduced and altered in several countries.

#### 7.2.4. Post-Harvest Sanitization

Sanitizing citrus fruits after harvesting reduces the risk of the spread of citrus canker through trade. For this purpose, various chemical sanitizers such as sodium hypochlorite, chlorine dioxide, peracetic acid, and calcium oxychloride are used [116].

#### 7.2.5. Systematic Acquired Resistance (SAR) Inducers

Systematic Acquired Resistance (SAR) is actually plant’s natural defense but can be activated in the absence of phytopathogens with the aid of chemical inducers [117,118]. It needs the signal molecule salicylic acid (SA) and is related to the accumulation of proteins related to pathogenesis [119].

Season-long control of foliar infection by Xcc is achieved by soil application of systemic neonicotinoid insecticides and the commercial systemic acquired resistance (SAR) inducer, acibenzolar-S-methyl (ASM). The protection provided by ASM is equivalent to that provided by copper hydroxide (CH) foliar sprays applied at 21-day intervals. When compared to the untreated control, all treatments dramatically reduced the occurrence of canker lesions on fruit. When started before the susceptible foliar flush in the spring, SAR inducers coupled with CH sprays offered the best control of fruit canker [120]. In the absence of high-intensity rains or tropical storms, ASM soil soaking and season-long rotations with thiomethaxom and imadocloprid were extremely efficient in controlling citrus bacterial canker on young grapefruit and orange plants. SAR treatments had a degree of control equivalent to eleven CH and/or STREP sprays spaced 21 days apart [121].

#### 7.2.6. Chemical Control of Citrus Leaf Miner (CLM)

Citrus leaf miner (*Phyllocnists citrella*) feeds on citrus leaves, thus facilitating the spread of citrus canker by exposing the leaf mesophyll tissues to the pathogen. Application of systematic neonicotinoid insecticides such as clothianidin and abamectin has been proven effective for its control [122,123]. The application of 1.5 g of the active ingredient (a 3:1 blend of (*Z*,*Z*,*E*)-7,11,13-hexadecatrienal–(*Z*,*Z*)-7,11-hexadecadienal) per hectare of citrus orchards in combination with permethrin also significantly eradicates its population [124,125].

### 7.3. Biological Control

#### 7.3.1. Genetic Engineering

The introduction of particular genes in plants and the overexpression of plant pattern-recognition receptors and exogenous defense-enhancing genes with the aid of genetic engineering techniques provide an innovative strategy to boost the plant defense system and disease resistance [126,127].

Invasion of a pathogen in the plant system triggers the activation of its defense system. Plants may synthesize antimicrobial proteins as a first line of defense, which includes a variety of small antimicrobial peptides [128,129]. Introductions of antimicrobial peptides into citrus rootstocks and scion cultivars have been reported to enhance their resistance to citrus canker. Thionins are cysteine-rich antimicrobial peptides whose modified form remarkably inhibited the pathogen growth and increased the plant resistance against citrus canker when overexpressed in Carrizo citrange [130]. The incorporation of antimicrobial peptide genes PR1aCB and AATCB in ‘Tarocco’ blood orange (*Citrus sinensis* Osbeck) inhibited the Xcc growth [131]. The integration of *Shiva A* and *Cecropin B* into the citrus genome has also been proven helpful in this regard [132]. Similarly, expression of the dermaseptin gene and attacin A gene from *Tricloplusia ni* in transgenic plants lessened their sensitivity to citrus canker. Attacins are antimicrobial peptides and are released into the insect haemolymph when the bacteria invade to cause infection. The genes encoding these antimicrobial proteins are isolated from specific insects and are used in genetic engineering in order to obtain resistance to phytopathogenic bacteria [133,134,135].

Studies have demonstrated that overexpression of AtNPR1 (Arabidopsis NPR1 gene), which is a key positive regulator of the systematic acquired resistance (SAR) in citrus plants, enhanced the resistance against citrus canker [136]. Greater expression of NbFLS2 (FLS2 gene from *Nicotiana benthamiana*) in transgenic citrus plants also increased its resistance against citrus canker and increased ROS production. The interaction of bacterial flagella and NbFLS2 improved plant basal defense. The development of canker lesions was also significantly reduced in some transgenic plants [126]. Increased expression of CsMAPK1 which is a citrus Mitogen-Activated Protein Kinase [137], and of the *Xanthomonas* resistant gene, Xa21, from rice, encodes for receptor kinase-like proteins [138,139] and can also be helpful in this regard. Fewer signs of epidermal rupture were observed in the plants with higher levels of CsMAPK1 [137].

Polyamines are thought to be a significant source of hydrogen peroxide production, which triggers the hypersensitive response and activation of defense-related genes. Thus, polyamines play a crucial role in establishing resistance in a plant against a specific disease. The endogenous polyamine level can be regulated by the overexpression of polyamine biosynthetic genes. Using this concept, a transgenic citrus plant was produced by the introduction of MdSPDS1, which is an apple spermidine synthase gene, into sweet orange (*Citrus sinensis* Osbeck ‘Anliucheng’). Overexpression of MdSDPS1 remarkably lowered the susceptibility of the transgenic plant to citrus canker [140,141].

Genome editing allows targeted genome modification for improving different traits of interest of many organisms and is garnering the attention of researchers, especially after the introduction of CRISPR-based systems (Clustered Regularly Interspaced Short Palindromic Repeats). So far, it is the most efficient and cost-effective genome editing technique [142]. The CsLOB1 (lateral organ boundary 1) gene of citrus plays a crucial role in disease susceptibility. The CsLOB1 promoter interacts with the pthA4-effector binding elements (EBE) to facilitate the attack of Xcc. CRISPR/Cas9-mediated genome editing of the CsLOB1 gene and the promoter remarkably reduced the susceptibility to citrus canker [143,144]. Reports have shown that CRISPR/Cas9-mediated genome editing of CsWRKY22, a marker gene for pathogen-triggered immunity in Wanjincheng oranges [145] and disruptive mutagenesis of CsDMR6 (Downy Mildew Resistance 6) in citrus varieties, [146] also have significantly reduced their susceptibility to citrus canker. CRISPER/Cas12a (Cpf1)-mediated genome editing [147] requires more attention as it exhibits higher efficiency and can be used as a powerful tool in future for the development of canker-resistant citrus varieties.

#### 7.3.2. Biological Control of CLM

The wasp *Ageniaspis citricola* is a natural predator of citrus leaf miner and thus helps in controlling its population and limiting the spread of citrus canker [148,149].

#### 7.3.3. Use of Plant Extracts

Plant extracts contain a variety of biomolecules that may possess antibacterial properties [150]. Thus, such extracts can be used as an alternative to environmentally unfriendly and costly synthetic bactericides. Sprays of various plant extracts on citrus plants have been proven to be essential in controlling Xcc, such as with extracts of *Allium cepa* L., *Calotropisgi gantea*, *Allium sativum* L., *Gardenia florida*, *Melia azedarch*, *Eucalyptus camelduensis*, *Azadirachta indica,* and *Dalbrgia sisso* [151]. Moreover, aqueous extracts of *Hibiscus subdariffa* L., *Punica granatum* L., *Spondias pinnata* L., and *Tamarinus indica* L lessened canker incidence in limes [152].

#### 7.3.4. Endophytic Bacteria

Endophytic organisms colonize the internal tissues of the host plants without causing harm to them [153]. Many endophytic bacteria act as biocontrol agents and positively affect the growth of host plant [154]. For biocontrol of citrus canker, many strains of endophytic bacteria such as *Burkholderia cepacia* [155], *Bacillus velezensis* [156], *Bacillus amyloliquefaciens* [157], *Kosakonia cowanii* [158], *Bacillus subtilis* [159], and *Bacillus thuringiensis* [160] have shown fruitful results. *Bacillus* strains have been proven very effective for canker control [159].

#### 7.3.5. Treatment with Bacteriophages

Different strains of Xcc, XauB, and XauC show sensitivity to various bacteriophages, which helps in reducing the inoculum [161]. Various Xcc strains isolated in Japan were found sensitive to Cp1 and Cp2 bacteriophages, whereas XauB strains were found sensitive to Cp3 bacteriophage (also mentioned in Table 2) [39,162,163]. A filamentous phage XacF1 was also found to be effective for canker control and has the ability to cause various physiological changes to the bacterial host cells [164]. Phage treatment can be more effective in combination with Systematic Acquired Resistance (SAR) inducers [165]. Treatment with bacteriophage combined with acibenzolar-*S*-methyl (ASM) significantly reduced the incidence of citrus canker [166].

## 8. Conclusions and Future Prospects

Citrus canker has been problematic for years due to high virulence, easy transmission, rapid spread, the complex pathogenic profile, mutations in the pathogen’s genome, and the presence of multiple strains. It can be predicted that new strains will continue to emerge in future. Even after eradication from a particular place, the risk of its reappearance remains. So, it is very crucial to develop a proper environmentally friendly strategy to completely control it. Genetic and molecular analyses have helped us in understanding its mechanism of action, which has paved the way for creating its control methods. More in-depth analyses of the pathogen and its evolving nature may help us find a key to a permanent solution. The use of endophytic bacteria as control agents also needs more research as it is an environmentally friendly method and shows great efficiency in controlling canker [159].

With the sequencing of the citrus genome and elucidation of its fascinating evolutive history [167], we now have unique tools to breed new varieties and look for desired traits in a targeted manner. Since traditional breeding of citrus varieties is challenging due to different constraints that include polyploidy, polyembryony, extended juvenility, and long crossing cycles, targeted genome editing technology has the potential to shorten varietal development for some traits, including disease resistance [144]. A great example of precise genome editing is the technique known as Clustered Regularly Interspaced Short Palindromic Repeats (CRISPR) and its associated Cas9 protein. Through this approach, it has been possible to confer resistance to pathogen infection in citrus by modifying an effector binding element in the promoter region of one single allele of the canker susceptibility gene CsLOB1 [168] or in both alleles of the gene in Duncan grapefruit [144]. Certainly, a long-lasting solution will be the development of citrus cultivars resistant to disease through engineering, and other strategies based on overexpression of pathogenesis-related protein PR1 have also shown some promising results [169].

## Figures and Tables

**Figure 1 plants-12-00123-f001:**
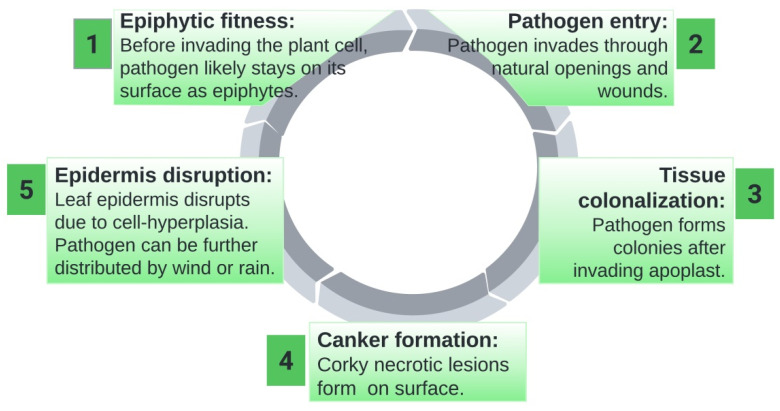
Disease Cycle of Citrus Canker.

**Figure 2 plants-12-00123-f002:**
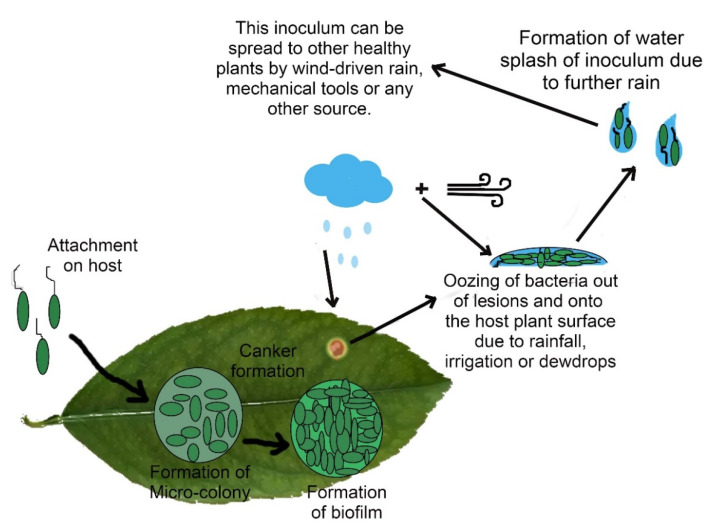
Stages of formation of biofilm and process of dissemination of inoculum.

**Table 1 plants-12-00123-t001:** Comparison of Pathovars.

	Pathotype					References
	A	B	C	D	E	
**Common disease name**	Asiatic canker	cancrosis B/false canker	Mexican lime cancrosis/cancrosis C	Citrus bacteriosis/Mexican bacteriosis(later named as citrus leaf spot)	Citrus Bacterial Spot (CBS)/Florida Nursery strain of CC	[34,35,36]
**Pathovar**	*citri*	*Aurantifolii* (Strain-B)	*aurantifolii* (Strain-C)	*aurantifolii* (Strain-D) (misunderstood)	*citrumelo*	
**Origin**	Asia	Argentina	Brazil	Mexico	Florida	
**Known geographical distribution**	Many citrus growing regions especially in Asia, USA, South America, Oceania	Argentina, Uruguay, Paraguay	State of Sao Paulo, Brazil	Mexico	Florida	
**Discovery**	1830 (controversial)	1923 on lemon	1963 on Key/Mexican lime	1981 on Key/Mexican lime	1984 on Swingle citrumelo	[6,34,37]
**Susceptible host**	Sweet orange, grapefruit, lemon, pummelo, mandarin, sweet lime, also observed in some other rutaceous plants	Lemons and Mexican lime, also observed in sweet orange, grapefruit, cider, mandarin, Volkamer lemon, sweet lime	Mexican lime, also detected in sour orange and lemon	Mexican lime	Swingle citrumelo, Grapefruit, mandarin, sour orange, sweet orange, lemon, Key/Mexican lime	[30,34,35,36]
** *pthA* ** **or its functional homologs**	Present	Present	Present	-	Absent	[30,31,38]
**Pathogenicity**	Highest	Low	High	Lower	lowest	
**Symptoms**	Distinctive corky necrotic lesions, sometimes possessing chlorotic or water-soaked haloes	Same as A, but symptoms take longer to appear, and lesions may vary in size from A	Similar to A	Similar to A	Flat water-soaked spots which may be surrounded by necrosis	[30,35]
**Parts of plant that may be affected**	Leaves, twigs, young stems, or fruits	Leaves, twigs, young stems, or fruits	Leaves, twigs, young stems, or fruits	Leaves, twigs, young stems, or fruits	Usually twigs and leaves only	
**Defoliation and dieback**	May occur	May occur	May occur	May occur	Does not occur	
**No. of Bacterial strains**	Many strains	Many strains	Many strains	Only one strain known (Xc 90)	Many strains	[39]

**Table 2 plants-12-00123-t002:** Comparison of major types of citrus canker.

		Xcc	XauB	XauC	References
**Host range**		Diverse	Less diverse	Restricted	[30]
**Xanthum gum production**		Highest	Almost 3 times less than Xcc	Almost 2 times less than Xcc	
**Cellular growth**		Non-fastidious, similar cellular mass values as XauC	Fastidious	Non-fastidious, similar cellular mass values as Xcc.	
**Comparative genetic analysis**					
**Genes related to flagellum synthesis**		Major 3 clusters of genes (F1, F2, and F3) present. Another 4th cluster is also present, consisting of 2 genes. Unrelated genes are also present in the region between F1 and F2 (XACSR9).	F2 gene cluster is absent. No genes are present in the region between F1 and F2.	All four gene clusters are present. No genes are present in the region between F1 and F2.	
**Presence of XacPNP gene**		Present	Absent	Absent	
**Type 1 Secretion System genes (T1SS)**		hlyB and hlyD encoding genes and TolC present	hlyB and hlyD encoding genes absent, TolC present	hlyB and hlyD encoding genes and TolC present	[31]
**Type 4 Secretion System genes (T4SS)**		Both in plasmid and chromosome	Only in plasmid, lack chromosomal copy	Only in plasmid, lack chromosomal copy	
**Genes involved in the regulation and synthesis of Type IV pilus (T4p)**		Many genes are present including pilX, pilV, pilA, pilL, and fimT, forming atleast 4 clusters of genes.	Among different clusters of genes, pilX, pilA, pilV.pil anf fimT genes are absent	Among different clusters of genes, pilX, pilA, pilV, pil, and fimT genes are absent	
**Genes related to synthesis of Hemagglutinin and Hemolysin**		Present in two regions of genome XAC4112-XAC4125 XAC1810-XAC1819	The genes present in 2nd region (XAC1810-XAC 1819) including fhaB and fhaC are absent	The genes present in 2nd region (XAC1810-XAC 1819) including fhaB and fhaC are absent	
**vapBC and tspO gene**		Present	Absent	Absent	
**Effector XopS**		Present (in some cases as pseudogene)	Absent	Absent	
**Effector XopK**		Present	Found as pseudogene	Found as pseudogene	[30,31,51]
**Effector XopF1, xopB, xopE4, xopJ, xopAF, xopAG**		Absent	Present (in some cases as pseudogene)	Present (in some cases as pseudogene)	
**Effectors xopE2, xopN, xopP, xopAE**		Present	Present	Absent	
**Basic physiological, biochemical, and serological tests**					[39]
**Reaction with Mab A1 (ELISA test)**		Reacts	Does not react	Does not react	
**Phage sensitivity**	**Bacteriophage Cp1**	Variable response	Insensitive	Insensitive	
	**Bacteriophage Cp2**	Variable response	Insensitive	Insensitive	
	**Bacteriophage Cp3**	Insensitive	Sensitive	Insensitive	
**Casein hydrolysis test**		Positive	Negative	Positive	
**Gelatin hydrolysis test**		Positive	Negative	Negative	
**Growth in presence of**	**3% NaCl**	Grows	No growth observed	No growth observed	
	**Maltose**	Grows	No growth observed	No growth observed	[54]
	**Aspartic acid**	Grows	No growth observed	No growth observed	
**General features of genome**					[15,30]
		Genome			
**Size (bp)**		5,274,174	4,877,808	5,012,633	
**# of contigs**		3	239	351	
**%GC**		64.7	64.9	64.8	
		Protein coding genes			
**Total**		4427	3804	3921	
**With functional assignment**		2779	2694	2728	
**Hypothetical**		262	117	184	
**Conserved hypothetical**		1386	993	1009	
		RNAs			
**rRNA operons**		2	2	2	
**tRNAs**		54	51	51	

XacPNP = Plant natriuretic peptide (belonging to Xcc); Mab = Monoclonal Antibodies; ELISA = Enzyme-linked immunosorbent assay; General features of genome of XauB and XauC are based on 94% and 96% of their genomes.

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
