# Peer review of "Citrus Canker Pathogen, Its Mechanism of Infection, Eradication, and Impacts"

_plants, 2022, doi:10.3390/plants12010123_

Round 1

Reviewer 1 Report

The submitted manuscript reviews citrus canker disease. It is stated in the first line that canker is the most devastating bacterial disease threatening citrus crops. Canker can be a serious disease, but the authors are overstating its global significance. Huanglongbing is universally considered to be the most devastating bacterial disease given its impact on the whole tree, and the difficulty of managing the disease.

In the manuscript, the authors have changed some of the familiar terminology relating to symptoms which may create confusion. For example, the term ‘bruises’ is used in the manuscript, but the literature typically refers to ‘lesions’ or ‘spots’, and ‘water-doused edge’ is more commonly known as the ‘water-soaked margin’. The use of the term ‘bruises’ is significantly misleading given the word bruise is commonly used to describe an impact injury on the fruit or other plant tissue. Why have twigs and fruit been separated into one paragraph, and stems into another? Aren’t twigs and stems the same thing? It would also be good to mention the significant impact that leaf miner can have on symptoms.

I suggest changing the language when referring to newer management strategies that are not used widely yet or proven in many situations – for example when describing SAR inducers you could soften the language from ‘control is achieved’ to ‘reports have shown a significant reduction in disease….’.

The English in the text needs to be improved prior to publication. There are many words used in the wrong context – I suggest using simpler words. There are also many errors – only some of which are outlined below.

Line 32 – Figure has no label – ‘Lesions’ is not spelled correctly, and the B in Biofilm is obstructed by the star.

Line 36 – suggest ‘… increase in attacks from many …’

Line 48 – delete ‘was’

Line 49 – citrus canker is not present in Australia – the 2018 outbreak was eradicated. It is mentioned later in the text that the disease was eradicated from Australia but the sentence stating that ‘canker is present in Australia’ is not correct.

Line 52 – replace ‘whereas’ with ‘and’

Line 53 – replace ‘another’ with ‘other’ - include a timeframe for China to provide perspective in relation to detections in India and Indonesia.

Lines 54-56 – how did the disease originate in all 3 disparate regions? Do you mean that canker could possibly have originated in one of the three?

Line 57 – ‘It is believed that citrus canker spread to other regions largely by movement of budwood.’

Line 65 – did citrus canker destroy the trees, or were the trees destroyed to eradicate citrus canker? Citrus canker has not been allowed to reach the level of a pandemic in Australia for decades. I suggest rewriting this whole paragraph to make it simpler. Where was the disease eradicated from? Where are they still trying to eradicate the disease? Where have they been unsuccessful and decided to stop trying to eradicate the disease?

Line 79 – replace ‘level’ with ‘scale’

Line 79 – ‘In a new review …’

Line 88 – c should be lower case in citri

Line 92 – define ‘It’

Line 95 – which Bureau of Plant Industry?

Table 1 – pathovar names should start with a lower-case letter

Line 116 – ‘Five major pathotypes …’

Line 118 – replace ‘further’ with ‘also’

Line 131 – delete ‘by’ and ‘analysis’

Line 135 – delete ‘analysis’

Line 138 – delete ‘The’

Line 140 – lower case a on the pathovar name aurantifolii

Line 183 – ‘… during the 1990’s …’

Line 221 – ‘… the appearance …’

Line 222 – ‘These early symptoms show as early as 4-7 days after the entry ..’

Line 227 – remove s from becomes

Line 232 – ‘… of the leaf …’

Line 236 – the word ‘blackened’ is misleading given the lesions do not change from white/yellow to black then back to light tan – maybe ‘The eruptions become stiff and corky, darkening in colour.’

Line 237 – it would be helpful to mention that most lesions on a leaf are of a similar size because there is typically only one infection period.

Line 242 – it would be helpful to mention why you don’t always see the halo around fruit lesions.

Line 259-260 – ‘Injuries become naturally contaminated at a lower rate than via the stomates …’

Line 268 – here you have used the word ‘bruise’ in a different context, replacing ‘bruise’ with ‘wounds’.

Figure 1 – should this be Figure 2? Under pathogen entry change to ‘Pathogen invades through …’. Correct the spelling of ‘hyperplasia’.

Line 280 – spelling of ‘adhesin’

Line 284 – ‘… is the TPS …’

Line 285 – ‘… to the TPS …’ and ‘… in the Xcc genome …’

Line 296 – ‘.. on a host …’ and ‘… of a host cell to reach the cell …’

Line 300 – ‘… has the ability …’

Line 320 – ‘… in the development of the bond …’

Line 321 – ‘… for successful …’

Figure 2 – correct the spelling of lesion

Line 347 – suggest changing to ‘wild type bacteria’

Line 348 – ‘conditions’ ‘plant cells’

Line 352 – delete ‘of’

Line 356 – ‘… of cell walls …’

Line 357 – ‘cells’

Line 360 – ‘… of introducing invasive plant pests and pathogens to crops.’

Lines 376-384 – I suggest stating that the example provided was in Florida

Line 387 – ‘… of the pathogen …’

Line 391 – ‘… of the pathogen …’

Line 392 – ‘cause’

Line 393 – ‘… facilitates bacterial …’

Line 395 – remove ‘been’

Line 398 – ‘high’

Line 397-401 – the practicality of this strategy is limited given the limited commercial potential of those varieties; it would be good to mention this.

Line 413 – ‘experiments’

Line 416-417 – ‘… they need to be sprayed …’

Line 418 – delete ‘its’

Line 424 – ‘experiments’

Line 425 – delete ‘been’

Line 450-452 – it is also important to mention community concern and regulatory issues with the use of antibiotics in some countries.

Line 489 – ‘… as a first …’

Line 491 – ‘… rootstocks and scion cultivars have been shown to enhance their canker tolerance or resistance…’ – both scions and rootstocks are varieties/cultivars.

Consider adding Parajuli et al. 2022 doi:10.1093/hr/uhac082 to the Genetic Engineering section.

Lines 526-529 – put scientific names in italics

Line 551 – delete ‘its’

Reviewer 2 Report

The authors systematically summarize the origins, types, symptoms and different prevention and control measures of citrus bacterial canker disease. This paper allows readers to quickly understand citrus canker disease and is a good review in the field of citrus bacterial canker disease.

Major:

The references in the '7.3.1 Genetic Engineering' section are too old to reflect the latest research advances in disease resistance gene mining and functional studies. In the past five years, there have been many new research advances in the prevention and treatment of citrus canker disease through genetic engineering, which need to be updated here.

Minor:

1. The legend of Figure 1 (line 32) was missing.

2. What exactly is the extract in '7.3.3 Use of plant extracts'? A brief description is needed about how these extracts is used.

Round 2

Reviewer 1 Report

The authors have not adequately responded to my correction that citrus canker is NOT present in Australia (as stated on line 50). All outbreaks in Australia have been declared eradicated. It is mentioned later in the text that the disease was eradicated from Australia but the sentence stating that ‘canker is present in Australia’ is not correct and therefore must not be published.

The authors have also not adequately responded to my comments about the paragraph describing eradication efforts. Citrus canker has not destroyed citriculture in Australia, it has not been allowed to reach that level and there has never been an incursion in a major growing region. There have been a number of incursions over the last century, all eradicated.

The authors have not adequately addressed the correction about their use of unfamiliar and confusing terminology relating to symptoms. For example, the term ‘bruises’ is used in the manuscript, but the literature typically refers to ‘lesions’ or ‘spots’, and ‘water-doused edge’ is more commonly known as the ‘water-soaked margin’. The use of the term ‘bruises’ is significantly misleading given the word bruise is commonly used to describe an impact injury on the fruit or other plant tissue. Twigs are stems and Section 5.3 mostly refers to all plants tissues therefore the heading should be changed to reflect the content.

The authors still have not defined adequately defined which Bureau of Plant Industry they are referring to – which country or state? The information about the growers doesn’t make sense to me – were the samples from there?

Line 124 – ‘… and the also …’ doesn’t make sense

Line 130-131 – ‘Disease associated by it …’ does not make sense

Line 132 – ‘m’ is not in italics and the rest of the name is in italics

Line 157 – correct the spelling of ‘orthologus’ to ‘orthologous’

Line 159 – ‘… greater in number than analogus number …’ does not make sense and analogous is misspelled

Line 231 – ‘… and temperatures in the range of …’

Paragraph 5.1 should be restructured to improve its flow. I suggest bringing the sentences describing the lesions earlier, then mention the shot hole effect, leaf drop and twig death at the end of the paragraph.

Line 280 – the spelling of ‘adhesin’ is incorrect – should be ‘Role of Adhesin proteins in bacterial adhesion’

Lines 299-300 – the sentence does not make sense

Line 309 – ‘… in the cell …’

Figure 2 – lesion is spelled incorrectly

Line 357 – suggest ‘… favourable conditions for its survival in plant cells …’

Line 366 – ‘… of host cells.’

Line 386 – ‘… all trees within a radius of 1900 feet (579 m) from an infected tree were removed …’

Line 388 – replace ‘lesser’ with ‘less’

7.1.1 – it needs to be clarified where this example occurred otherwise it is confusing for the reader.

Lines 395 and 399 – ‘… of the pathogen …’

Line 403 – change to ‘… has proven effective …’ – you should not use the word ‘more’ without defining what you are saying something is ‘more than’.

Line 406 – replace ‘higher’ with ‘high’

Line 409 – explain why the practicality of the strategy is limited

Line 407 – replace ‘higher’ with ‘more’

Line 431 – replace ‘experimentations’ with ‘experiments’

Line 585 – delete ‘its’

Line 587 – delete ‘there’

Lines 588 and 592 – ‘environmentally friendly’

Line 602 – ‘… confer resistance to pathogen infection’ – it is the pathogen that infects, not the canker
